# AvaLife—A New Multi-Disciplinary Approach Supported by Accident and Field Test Data to Optimize Survival Chances in Rescue and First Aid of Avalanche Patients

**DOI:** 10.3390/ijerph19095257

**Published:** 2022-04-26

**Authors:** Manuel Genswein, Darryl Macias, Scott McIntosh, Ingrid Reiweger, Audun Hetland, Peter Paal

**Affiliations:** 1MountainSafety.info, 7260 Davos, Switzerland; 2Department of Emergency Medicine, University of New Mexico, International Mountain Medicine Center, Albuquerque, NM 87131, USA; dmacias@salud.unm.edu; 3Department of Emergency Medicine, University of Utah Health, AirMed, Salt Lake City, UT 84132, USA; scott.mcintosh@hsc.utah.edu; 4Institute of Mountain Risk Engineering, University of Natural Resources and Life Sciences, 1190 Vienna, Austria; ingrid.reiweger@boku.ac.at; 5CARE Center for Avalanche Research and Education, UiT The Arctic University of Norway, 9010 Tromsø, Norway; audun.hetland@uit.no; 6Department of Anesthesiology and Intensive Care Medicine, St. John of God Hospital, Paracelsus Medical University, 5010 Salzburg, Austria; peter.paal@icloud.com

**Keywords:** hypothermia, accidental, asphyxia, avalanche, basic life support, cardiac arrest, emergency medical services, mountain rescue

## Abstract

Snow sports in the backcountry have seen a steep increase in popularity, and therefore preparedness for efficient companion and organized rescue is important. While technical rescue skills are widely taught, there is a lack of knowledge regarding first aid for avalanche patients. The stressful and time-critical situation for first responders requires a rule-based decision support tool. AvaLife has been designed from scratch, applying mathematical and statistical approaches including Monte Carlo simulations. New analysis of retrospective data and large prospective field test datasets were used to develop evidence-based algorithms exclusively for the avalanche rescue environment. AvaLife differs from other algorithms as it is not just a general-purpose CPR algorithm which has been slightly adapted for the avalanche patient. The sequence of actions, inclusion of the ≥150 cm burial depth triage criterion, advice to limit CPR duration for normothermic patients to 6 min in case of multiple burials and shortage of resources, criteria for using recovered subjects as a resource in the ongoing rescue, the adapted definition of “injuries incompatible with life”, reasoning behind the utmost importance of rescue breaths, as well as the updated BLS-iCPR algorithm make AvaLife useful in single and multiple burial rescue. AvaLife is available as a companion rescue basic life support (BLS) version for the recreational user and an advanced companion and organized rescue BLS version for guides, ski patrols and mountain rescuers. AvaLife allows seamless interoperability with advanced life support (ALS) qualified medical personnel arriving on site.

## 1. Introduction

In avalanche rescue, time is critical for survival. Standardized first aid is required for a better outcome for avalanche patients. An early prototype of AvaLife was published in a first, single-author publication in 2013 [1]. Since then, it has undergone a peer-reviewed verification of specific sub-aspects, such as the feasibility of using Monte Carlo simulations for this specific application [2]. With more than one buried subject, a shortage of resources is likely, in particular during the first phase of companion and organized rescue. The concept of remote reverse triage in avalanche rescue has already been introduced in 2008 [3]. In remote reverse triage, sometimes referred to as rescue triage, a first step of triage is made in the search and rescue phase, based on parameters such as terrain characteristics, vegetation and burial depth. Excavating the buried subject represents the by far most time-consuming and resource-absorbing effort of avalanche rescue. Therefore, prioritizing buried subjects, who require less excavation effort and were exposed to less mechanical impact while transported downslope in the avalanche, leads to a more favorable overall outcome.

In 2010, Bogle and Boyd, working for Canada’s helicopter skiing industry, introduced AvSORT I [4], an avalanche patient triage algorithm focusing on the specific needs of the investigators’ operational environment, an uncultivated landscape where skiing activities are very often carried out in forested terrain. In the Canadian Rocky Mountains, traumatic cardiac arrest is more frequent compared to Europe (27%, compared to 6%) [5]. Furthermore, in British Columbia, performing on-site advanced life support (ALS) measures is very unlikely; additionally, transport distances to extracorporeal life support (ECLS) facilities are much further compared to the European Alps. In 2021, an updated AvSORT II version was presented at the International Society of Mountain Medicine (ISMM) 2021 congress (https://ismm2021.org as of 30 June 2021), using the START triage algorithm. This algorithm does not allow for medical treatment to be rendered to patients with absent vital signs during the phase of reverse triage. For a basic life support (BLS) provider, the inability to obtain vital signs in the field despite being present has been demonstrated to be unreliable, with a potentially 50% false negative rate. In 2013, the International Commission for Mountain Emergency Medicine (ICAR MEDCOM) published the Avalanche Patient Resuscitation Checklist for the treatment of a single avalanche patient [6]. An update to the ERC 2015 avalanche rescue guidelines was published in 2017 [7]. The checklist is also used for the situation where there are sufficient resources available to treat all patients with the highest level of care. The protocol includes neither rescue tactical considerations, nor recommendations for reverse triage situations. Therefore, in a resource-limited situation, the ICAR MEDCOM checklist provides no tactical options during search and excavation. It also excludes tactical options for BLS-trained first aid providers, and limits ALS tactical options to achieve the “greatest good for the greatest number” in terms of survival.

According to European data, survival chances drop by 70% within the first 35 min of burial duration, thus a decrease of 2% per minute [5]. Taking only the period from 15 to 35 min into account, the decrease is almost 3% per minute. This dramatic reduction in survival shows that avalanche accidents are amongst the most urgent emergencies in mountain rescue because every single minute counts. Of all avalanche victims in Canada, 81.1% die due to cardiac arrest from asphyxia, 18.9% due to traumatic cardiac arrest [5]. In Europe, the causes of death are similar, with few succumbing to primary hypothermic cardiac arrest [8]. Traumatic cardiac arrest is the by far most unlikely cause to be survivable in a mountain environment (0.8–1.2% for non-mountain related accidents) [9], and treatment options for BLS-trained rescuers are very limited. A lack of oxygen is responsible for the 70% suffering from asphyxia leading to cardiac arrest, mostly within the first 60 min of burial. Hypothermic cardiac arrest tends to occur after 60 min of burial duration, which deserves closer attention [10,11].

The goal of AvaLife is to offer useful, evidence-based search and rescue and medical treatment algorithms in one avalanche accident-specific protocol. AvaLife shall be applicable by BLS qualified users and be compatible to the very diverse situations in mountain ranges all over the world. The aim of this study is to present the current version of AvaLife for BLS rescuers (Figure 1).

## 2. Materials and Methods

This was an experimental observational field study without human subjects; it did not require any institutional review board review or clearance. Analysis of the avalanche rescue problem starts at the beginning of the search and rescue operation, without depending on the initial organizational form of the rescue effort; it is thus employed the same way for companion and organized rescue. In all AvaLife modules, several tasks should be performed in parallel, if the available resources permit. However, in case of limited resources, AvaLife will initially prioritize limited resources towards buried subjects and patients whose rescue requires lower resources while simultaneously focusing on subjects with a statistically higher chance of survival.

Concerning the development of the “search and excavate” module, optimizing overall survival chances in a low-resource setting necessitates the identification of subjects who (a) had a lower potential of severe mechanical impact while being transported downslope in the avalanche, (b) are on the surface or as closely as possible to the surface and (c) require minimal excavation effort. The relevant terrain characteristics have been identified in the publication “Remote reverse triage in avalanche rescue” [3].

Concerning burial depths, the time required to excavate a buried subject often exceeds the time required to make another find. Therefore, the potential to optimize survival chance by postponing the excavation of particularly deeply buried subjects needed to be systematically analyzed and cut-off points for cases where there is a shortage of resources needed to be determined.

A dataset including 1070 full burials from the accident databases from Austria (1 November 2005–31 October 2021), Switzerland (1 November 1999–29 October 2021) and France (1 November 1999–31 October 2019) has been used. Cases with a high uncertainty on burial depth, burial times or the search mean applied have been excluded. Excluded were all accidents on roads, railroad and residential areas (Urban Avalanche Search and Rescue UAvSAR), cases where the search mean is unknown as well as cases with a low confidence rating concerning burial duration and burial depth.

In a first step, we calculated the mean burial depth of excavated subjects for multiple ranges of burial duration based on the 1070 full burial dataset from France, Austria and Switzerland. This data is required to connect the survival chance derived from the Haegeli et al. survival curve to the mean burial depth of a specific burial duration range [5]. None of the existing survival curves for fully buried subjects differentiate between different burial depths; therefore, the probability of survival for any given burial duration always includes buried subjects at different burial depths. However, as excavation performance by rescuers is similar and mechanical properties of debris of different avalanches equally have similarities, mean values of burial depths of excavated patients could not be assumed as equal for all burial durations.

In a second step, we calculated the decrease in probability of survival due to burial depth. In order to clearly distinguish between the impact of burial duration and burial depth to mortality, the probability of survival for small ranges of burial depth and small ranges of burial duration were calculated separately. To analyze potential reasons for a change in mortality in greater burial depth, cause of death and the presence of an air pocket have been analyzed.

Applying a Monte Carlo simulation, we evaluated the threshold for the highest probability of survival, above which excavation should be postponed in case of multiple burials and a shortage of resources. With respect to search duration for the first buried subject, we simulated 5, 10, 25 and 40 min scenarios. These values include the time for the rescuers to arrive on the scene. These specific durations have been selected to simulate two different companion rescue scenarios (5 and 10 min), a scenario of fast-arriving organized rescuers (25 min) and an organized rescue scenario with a later arrival time on the scene (40 min). Mean search duration until probe hit for the first buried subject in advanced companion rescue simulation is 3:43 min. Due to less mental preparedness and additional stress in case of a real accident, a 25% search duration has been added in the considerations for the 5 min companion rescue scenario. As the duration to the first find depends on various variables with the potential to lead to a considerable increase in search time, in particular the position of the members of the party who have not been caught by the avalanche relative to the deposit area, 10 min has been chosen for the second companion rescue scenario. In organized rescue, mean arrival times of approximately 20 min are only recorded for densely populated areas with high mountain tourism. Ski patrols act in a similar timeframe or are slightly faster. Taking 5 min for the search into account leads to the 25 min scenario. The 40 min scenario is more realistic outside of the core regions of mountain tourism, but still represents an interesting scenario compared to intervention times over 60 min, where survival chances are fairly low and decrease only slowly. Medical treatment time for the first patient was set to 10 min, which is in alignment with the suggested cardiopulmonary resuscitation (CPR) duration in case of resource shortages (including 4 min of search and extrication and 6 min of CPR). Search duration for the second buried subject has been varied between 5 and 10 min. Burial depths for patients 1 and 2 were randomly chosen from the Swiss accident database. For each combination, we calculated the overall survival chance for the scenario where the first buried subject is excavated immediately, and for the scenario when the rescuer decides to postpone excavation of the first buried subject, moving on to search and excavate the second buried subject before returning to the first. In the case that the second buried subject was found to be considerably deeper than the first, we calculated the effort to return to the first subject before excavating the second subject. In all scenarios, both buried subjects were excavated, and their cumulative probability of survival represents the overall life-saving performance measure. To calculate the survival chance of a buried subject, we derived the probability of survival from the Haegeli et al. survival curve [5] and then applied the burial depth-related mortality correction factor based on the difference between the real burial depth and the mean burial depth of the respective burial duration range.

We equally investigated the impact of the hardness of the debris. In hard debris, rescuers remove a median depth of 13.20 cm/min; in soft debris, a median depth of 25.32 cm/min is removed. Excavation performance has been measured by field testing in four nations, with multiple user groups and in debris of different hardness, resulting in a total of 391 rescuers applying the snow conveyor belt excavation technique. Hard debris is considered equal to a snow profile hardness test of 1 finger or harder, and 4 fingers or softer for soft debris as. The hardness rating scale of knife, pencil, one finger, four fingers and fist is internationally applied and standardized in snow science and avalanche forecasting [12]. It is of utmost importance to realize that the density of debris is only making a very marginal impact compared to the hardness of the debris. Thus, a low cohesion, but high-density mass of snow only marginally increases excavation duration, whereas hard debris immediately leads to an important increase in excavation duration.

To determine the threshold for the highest probability of survival for a given burial depth, we tested threshold depths between 100 and 300 cm in 10 cm increments. For each step, the Monte Carlo simulation tested 1000 cases, so as to make the influence of an unrepresentative distribution of accident and burial characteristics negligible.

In order to quantify a threshold for the criterion of “shortage of resources present”, and thus “reverse triage mode applies”, the minimum number of required rescuers for an efficient excavation effort needed to be determined. The required length of the snow conveyor belt depends on burial depth and slope inclination. We calculated the median burial depth based on the dataset of 1070 buried subjects from Austria, Switzerland and France. To evaluate burial site inclination, a dataset of 800 avalanches of the type “avalanche accident” has been taken into account (provided by the Institute for Snow and Avalanche Research SLF, Switzerland), and inclination values were calculated by the Rapid Mass Movement Simulation (RAMMS) of the SLF, a department of the Swiss Federal Institute for Forest, Snow and Landscape Research WSL. All pixels within the outline of the debris with a calculated deposition depth of greater or equal to 0.5 m have been considered, which has led to a total of 38,781 datapoints. In parallel, employees of 10 avalanche forecasting centers and leading avalanche experts from Europe and North America have been interviewed on the question of “average slope angle of the excavation site”.

Medical treatment is initiated as quickly as possible, since every additional minute reduces survival chance by 2 to 3% for 58% of buried subjects in our database who were excavated within the first 35 min after an avalanche. Therefore, we had to identify the priorities and define the procedures for the last phase of excavation in close proximity to the patient. Once the first medical assessment is possible, two new fields of interest open up in the case of multiple buried subjects present during a shortage of resources. First, does the patient require immediate medical treatment—and might it even be possible to use this first patient to assist in the ongoing rescue? In the second case where the initial assessment shows that the patient is not awake, does not give clear answers, or is injured, we evaluated in which chronological sequence further medical assessments and potential life-saving treatments should be performed. Special attention has been paid to the fact that the vast majority of buried subjects quickly develop severe respiratory depression. This led to an evaluation of the most suitable interventions for asphyxiating patients, a process matched with a drowning patient, which is the closest proxy to the avalanche patient. Drowning accidents are much more frequent globally compared to avalanche accidents; thus, much more research has been invested in the research and guidelines for the treatment of drowning patients. Given that the pathophysiology of asphyxia in drowning and avalanche burials approximate one another, our results and conclusions for an optimal treatment of a suffocating avalanche patient closely follow the conclusions of the guidelines for drowning patients [13].

To ensure that the AvaLife algorithm reflects the practical sequence of actions in a real accident case, we analyzed the last phase of excavation, and the chronologically overlapping first phase of medical treatment. The goal of a short duration until the head is accessed implies that the width of the snow conveyor belt has to be limited to approximately 150 cm. As soon as the patient becomes visible, the tip of the snow conveyor belt is enlarged as quickly as possible, directed towards the head and chest of the patient. Due to the limited width of the snow conveyor belt, a maximum of two rescuers can work in immediate proximity to the patient. Therefore, it is impossible to free the entire body of the patient simultaneously; all efforts must be focused on head and chest access, to initiate airway, oxygenation, ventilation, and CPR strategies. In contrast to treatment guidelines for non-buried patients, we concluded that the first potentially life-saving measures must be initiated before asking oneself whether the patient has “injuries incompatible with life”. As soon as first aid has started, the second rescuer will focus on freeing the rest of the body, allowing a proper body check and assessment of “injuries incompatible with life”.

The analysis of the optimal chances of survival with respect to CPR duration for normothermic patients during limited resources has been performed based on a Monte Carlo simulation [2].

For the intermittent CPR (iCPR) module of the BLS version of AvaLife, it was necessary to develop a set of rules to ensure that the precondition of a core temperature <28 °C is met. For every 1 °C cooling, oxygen consumption falls by 7% [14]. However, only below 28 °C is there sufficient certainty [8] that the 5 min of CPR interruption after 5 min of regular CPR will not cause hypoxic brain damage [15]. Initial suggestions have combined the criteria of burial duration >60 min with a subjective estimation of temperature and the potential for severe heat loss: “chest feels cold AND burial duration allowed for severe heat loss (clothing/insulation)”. We did not consider the absolute accuracy of a surface temperature estimation to be sufficiently reliable, since this estimate is made by BLS rescuers exposed to the harsh mountain environment. Both conditions lead to a further deterioration of temperature-sensing capabilities, which should not be considered to provide absolute accuracy even in normal conditions but only approximate relative accuracy. It is important to note in this context that the >60 min criterion for treatment as a potential hypothermic cardiac arrest patient (core temperature < 30 °C) is only met with seldomly reached cooling gradients [9] of up to 9 °C/h [10,15]. To avoid critical undertreatment, we assume that the avalanche patient has suffered hypothermic cardiac arrest at 30 °C and only immediately before the rescuers have reached the head and chest of the patient. The assumed cooling gradient is therefore 7 °C ÷ burial duration in hours. To reach the threshold of <28 °C to qualify for iCPR, the patient needs to cool down another 2 °C. Assuming a linear cooling gradient during burial duration and CPR duration outside of the avalanche, cooling down another 2 °C will require an additional duration of: burial duration [h] ÷ (7 °C ÷ 2 °C). To simplify the calculation and add a safety margin for imprecisely determined burial duration, “burial duration ÷ 3” instead of “burial duration ÷ 3.5” is applied. A further, very considerable safety margin is given by the exposure of the patient receiving CPR to outside temperature, potential windchill and strongly increased heat loss induced by the homogenization of the body temperature due to regaining blood flow and thus transport of warmer blood to superficial layers.

## 3. Results

Optimizing survival chance during search and excavation necessitates directing resources first to non- and partially buried subjects, with the additional benefit that they might provide important information about the total number of people missing, and available equipment. The search effort for the completely buried subjects with no visible parts prioritizes zones with a lower probability of severe mechanical impact during the dynamic phase of being transported downslope as part of the debris flow. Consequently, zones with a high probably for high-energy impact due to fall or collision need to be excluded in the first place (reverse remote triage). Once the rescuers have arrived on the surface of the debris at the point where the remaining distance to the buried subject is as short as possible, burial depth can be estimated or precisely determined by probing.

Mean burial depth of subjects excavated in different ranges of burial duration shows a linear increase in burial depth over burial duration (Figure 2).

Our data show that buried subjects excavated from deeper burial depths were less likely to survive than buried subjects in shallower burial depths, despite the equal burial duration. The decrease in survival probability percentages per meter of burial depth shows its greatest impact in the first 40 min of burial duration (Figure 3). Therefore, companion and fast organized rescuers need to pay particular attention to these facts when dealing with multiple buried subjects during a shortage of resources. The mean burial duration in our dataset of 1070 patients is 30 min. A total of 42% of the subjects were excavated within the first 20 min, 62% within the first 40 min and 71% within the first 60 min after the avalanche occurred. Almost two-thirds of the subjects fall within the burial duration range, where burial depth has a particularly important impact on survival chances. The longer the burial duration, the lower the survival chance influenced by burial depth because shallow burials show an almost equally high mortality based on the single factor of prolonged burial duration (Figure 3 and Figure 4). A detailed analysis of the reasons behind the increase in mortality in increased burial depths is not part of this publication. However, our data indicates that the percentage of deaths with trauma as cause of death decreases with increasing burial depth (25% within the first meter of burial depth, 17% within the second meter of burial depth and 5% within the third meter of burial depth). At the same time, the median burial depth of patients without an air pocket is 100 cm, considerably deeper compared to 60 cm with an air pocket.

The Monte Carlo simulation shows the highest number of lives saved with a burial depth threshold of very close to 150 cm independently of the search duration for buried subject 1 (including intervention duration) and independently of the hardness of the debris. In the scenario of swift companion rescue with 5 min to find the first buried subject, the difference in survival chance between hard and soft debris is 14.5%, with increasing search duration in scenarios 2 to 4; as expected, this difference steadily decreases to 9.7% for scenario 2, 3.5% for scenario 3 and 1.8% for scenario 4 (Table 1).

The median burial depth of the 1070 buried subjects from Austria, Switzerland and France is 100 cm, a result that is in line with previous results from multiple nations. The RAMMS simulation showed a median inclination of the excavation sites of 7.7°. The result of the expert-level consensus is a slope angle of 12.5°, which fits reasonably well with the quantitative result with a median value of 7.7° and a mean value of 11.2°. Applying the rules for the required length (Figure 5) and required number of rescuers (Figure 6) needed for excavating a buried subject applying the snow conveyor belt technique, two rescuers are required (Figure 7) and sufficient for 7.7°.

To initiate the first medical treatment as quickly as possible, the first priority of the excavation effort is freeing the head and chest of the patient. Therefore, the interface between the end of the search and rescue effort and the start of the medical first aid is dynamic and overlapping in time. As every minute counts, withholding first aid, to completely free the patient with all extremities would lead to a considerable decrease in survival chances.

In case of a shortage of resources, the Out-Of-Hospital Medical Treatment module (Figure 8) applies a first filter, postponing the complete excavation and treatment of patients who are conscious and do not show signs of life-threatening injuries. For this specific subgroup of patients, we have defined the criteria under which such people might be included as rescuers in the ongoing rescue effort. The considerations include on one hand the potential of the person to substantially support the rescue effort, and, on the other hand, estimate the effort required to completely free the person and prepare them to become part of the rescue team. An adequate level of monitoring the condition of people who were initially caught by the avalanche and then assist in the rescue effort is an integral part of the above-mentioned considerations.

For all avalanche patients needing immediate care, first priority is given to airway status and clearing the airway if required. This is immediately followed by checking if the breathing of the patient is normal. Where the patient is not breathing normally, five rescue breaths should immediately be applied [13]. AvaLife is the first medical treatment algorithm for avalanche patients which fully recognized that this group of patients experience severe respiratory suppression with the potential to quickly escalate into a hypoxic cardiac arrest, and thus advises following the European Resuscitation Council (ERC) recommendations for drowning patients [9]. These recommendations have been adopted as well in the ERC 2021 avalanche rescue algorithm and guidelines [13].

The majority of patients are buried face down in the avalanche [17], therefore turning the patient into a position allowing rescue breaths is required. If vital signs can be detected because of rescue breaths, further treatment of the patient should be continued following general wilderness first aid protocols, primarily focusing on treating life-threatening injuries and conditions. However, the sole presence of agonal breathing should not be interpreted as a sign of life [18].

Patients who do not show vital signs after five rescue breaths need to be resuscitated applying the 30:2 cardio-pulmonary resuscitation protocol [13,19]. If there is no shortage of resources, thus all other buried subjects are currently being excavated or have already received medical treatment, CPR should be applied by BLS-trained rescuers until the patient can be handed over to professional health care providers. Without shortage of resources, BLS-trained rescuers may only terminate resuscitation if their own life is at an unacceptable level of risk or if there are body features incompatible with life [20,21].

In case of a shortage of resources, all rescuers, independently of if they are BLS or ALS qualified, have to follow the “Greatest good for the greatest number” paradigm in order to make sure that as many lives as possible can be saved with the limited resources available. This requires focusing on patients with higher chances of survival and implies that treatment of other patients may be temporarily postponed.

For patients with injuries incompatible with life, this means that further treatment will be postponed. While CPR needs to start as soon as possible, the lower extremities might still need to be fully freed and snow sports equipment removed. It might take some time until there is sufficient space around the patient allowing proper examination of the extent of the mechanical impact potentially suffered during the dynamic phase of being transported downslope in the debris flow. Where there is a shortage of resources, the resuscitation efforts of normothermic patients should be limited to three treatment windows of two minutes each, thus six minutes in total. As shown in a Monte Carlo simulation, specifically focusing on CPR duration in light of providing ‘Greatest good for the greatest number’, overall survival chances decrease where six minutes of CPR is exceeded [2].

For the potentially hypothermic patient with a burial duration greater than 60 min, survival is only possible if the airways were not fully obstructed [13]. If at least some gas exchange was possible, or the condition of the airways on reaching the patient is unknown, CPR with (theoretically) unlimited treatment duration is advised. The procedure to check for injuries incompatible with life is carried out in the same manner as described in the section of the normothermic patient.

For all hypothermic patients, only up to three defibrillations are recommended [22], as it is likely that the low core temperature does not lead to successful defibrillation with a sustainable effect. Furthermore, repetitive use of defibrillations may cause unnecessary myocardial injury. As automated external defibrillators (AED) do not include technical diagnostic measures to exclude this type of misuse, this limitation and exception from regular BLS AED guidelines is specifically mentioned.

In case avalanche patients in cardiac arrest with a core temperature < 28 °C need to be transported under conditions not allowing efficient CPR due to extreme shortage of resources, intermittent CPR (iCPR) may be considered (Figure 9) [15].

Before iCPR may be applied, the BLS rescuer must first apply uninterrupted CPR for 1/3 of the burial duration.

The staging and treatment of hypothermia is defined in the Hypothermia Staging module (Figure 10) of AvaLife and is based on the Revised Swiss System [23,24]. Shivering has been removed as a decisive criterion to determine hypothermia stage since it cannot be generalized and is furthermore influenced by age and traumatic impact [23]. The state of consciousness based on the Alert, Voice, Pain, Unresponsive (AVPU) scale can be used to derive hypothermia stage, and therefore the risk of a hypothermic cardiac arrest and the appropriate counter measures. While active rewarming by moving is appropriate in hypothermia stage 1, slowing down further cooling by insulation, heat packs and separation from cooling snow is the priority in hypothermia stages 2 and 3. Warm, sugary drinks are recommended in hypothermia stages 1 and 2 and have the main objective to regain metabolism. For patients in hypothermia stage 2, consider giving something to drink only if the patient is able to properly swallow and the upper body can be slightly inclined.

In hypothermia stage 4, where there are no signs of life, the priority is effective CPR. Effective CPR renders many of the measures to avoid further heat loss impracticable. However, as manual BLS CPR is unlikely to provide a high-quality perfusion over a longer period of time, further reduced oxygen consumption due to the decreasing core temperature should be seen as an overall advantage to reduce the potential of hypoxic brain damage [22]. If hypothermic cardiac arrest occurs in a patient who is severely injured, a further decrease in core temperature during resuscitation is likely to lead to further complications due to the negative impact on coagulation. Unfortunately, such combinations show a low probability of survival if there are still only BLS qualified rescuers on site after 60 min of burial duration.

## 4. Discussion

The first ever analysis on the impact of burial depth on survival chances has shown an important decrease in survival chance for every additional meter of depth in the first 40 min of burial duration. To our surprise, the output of the Monte Carlo simulation clearly shows that the survival chance-optimized burial depth threshold is equal for soft and hard debris, even though excavation performance in soft debris is almost twice that in hard debris. However, the extent of the decrease in survival chance due to burial depth is so dominant that even a solid excavation performance of 25.32 cm/min did not lead to a deeper burial depth threshold for soft debris.

Due to the very limited treatment options on the avalanche, the life of very severely injured patients is most often unsalvageable, even after short burial durations and in shallow burial depth. At the same time, the likelihood of a BLS-trained companion or organized rescuer being able to reverse the effect of an increasingly asphyxiating patient who is not in cardiac arrest is high. Therefore, the shorter the burial duration, the higher the survival chance of patients who would otherwise later die of hypoxia. The likelihood of an air pocket decreases with increasing burial depth, indicating that the average size of an air pocket also decreases with increasing burial depth. This has the potential to dramatically speed up the onset of asphyxia and hypoxia, and therefore shortens the duration time cardiac arrest despite very efficient excavation to overcome the problem. 

The outcome of the Monte Carlo simulation shows only a very marginal (+max. 0.023 mean lives saved) advantage of situation-specific burial depth thresholds. Therefore, the advantage of a single threshold solution is overwhelming. By coincidence, the burial depth threshold of 150 cm is equal to the probing depth in the first passage of a probe line, another field of interest where Monte Carlo simulations have been applied to optimize the probability of detection versus survival chances. When it comes to depth-related considerations in avalanche rescue, 150 cm is a key value to remember!

The fully quantitative and evidence-based criterion when immediate excavation should be started independently of burial depths is “two or more rescuers per remaining buried subject”. This outcome confirms the previous recommendation, which was only backed by some simplistic quantitative considerations. Considering that the required number of rescuers is decreasing with an increasing slope angle, steeper excavation sites are in favor of the situation as less debris will need to be removed (Figure 5). The 20% percentile of slope inclination of burial sites is at 3° and leads to 90 cm segments instead of 80 cm per rescuer (Figure 5, burial depth 100 cm, length = tan (90° − 26° − 3°) –> 180 cm -> 180 cm/2 rescuers = 90 cm); this shows that most cases can be properly handled with almost no influence on excavation efficiency.

An important difference to the chest compression-only recommendation taught in most BLS AED courses is the absolute requirement to give rescue breaths. The AvaLife footnote “No rescue breaths –> no oxygen –> no return of spontaneous circulation –> no survival.” will clarify the ultimate importance of rescue breath for the suffocating avalanche patient.

For AvaLife, we developed an avalanche-specific definition of “injuries incompatible with life”. This was necessary as the type of mechanical impact caused by being transported downhill in a snow avalanche, including potential collisions with rock and trees, typically leads to blunt trauma. The filter effect of the definition “obvious lethal injuries”: decapitation, truncal transection, or whole body frozen is therefore ineffective. Filters without filtering effect due to threshold criteria which are not reached in the extent or type of injury are ineffective and thus have to be avoided. Based on the avalanche specific mechanical impact pattern, we have defined “injuries incompatible with life” as follows: Obvious evidence of severe mechanical impact/high fall/severe collision with trees or rocks or a head/truncal body position incompatible with life (Figure 8).

The rationale behind the 6 min CPR duration for normothermic patients ensures that the “Greatest good for the greatest number” paradigm is met in case of a shortage of resources. While it is important to take advantage of the fact that the likelihood of reaching return of spontaneous circulation (ROSC) is highest in the first few minutes of CPR, the overall life-saving effect quickly becomes negative as the ROSC curve flattens out and the rapidly decreasing survival chances of those who are still waiting to be excavated become a greater toll than the benefit of prolonging CPR. The advice to limit CPR to six minutes in this particular situation is further supported by the recommendations of the ERC 2021 guidelines that ‘The chance of a good outcome is improved if there is a ROSC in the first minutes of CPR” [13]. While the duration for BLS is (theoretically) unlimited if sufficient resources are available, the required benefit to justify the investment of the spare resources in a reverse triage situation is substantially higher as it needs to outweigh the loss of survival chances of those who are not getting any treatment at all.

Even though some might see the BLS iCPR algorithm starting with a non-interrupted CPR duration of 1/3 of burial time as a burden, there is no way around it without a reliable core temperature measurement. The BLS iCPR algorithm ensures that: (a) patients in hypothermic cardiac arrest only receive iCPR once they have reached a core temperature < 28 °C, and (b) patients in normothermic cardiac arrest receive at least 25 min of uninterrupted CPR before the first 5 min interruption of the iCPR protocol takes place. This combination only exists with a burial duration immediately after reaching the >60 min criterion. With longer burial duration, the duration of uninterrupted CPR increases. These considerations are only relevant to normothermic patients who have arrested within the last 1–4 min before reaching the head and chest of the patient, all other patients who have previously suffered normothermic cardiac arrest are unsalvageable. Note that the iCPR protocol always starts with at least 5 min of uninterrupted CPR before the first interruption of a maximum of 5 min may be applied.

This study has several limitations: The AvaLife project is an experimental study. Even though several countries have introduced AvaLife as their new avalanche patient treatment protocol, there is only a low number of real-life rescues where it has been operationally applied. Designing a prospective, blinded, case–control study would likely be unethical. However, AvaLife has been presented and taught in practical workshops in 23 countries during the past 8 years. The hundreds of training sessions, indoor lectures and webinars including AvaLife, as well as translation to 17 languages to date, have led to a multi-year fine-tuning process based on feedback from 27 countries. The results of this work are presented in this study.

## 5. Conclusions

AvaLife provides a systematic, flowchart-based approach for search and rescue as well as the out-of-hospital treatment of avalanche patients. With its four different modules in the BLS version, it is a comprehensive decision-making support tool that can also be used as a protocol to collect the most critical patient data as a base for further in-hospital treatment.

In contrast to other tools and algorithms for the medical treatment of avalanche patients, AvaLife is not just a general-purpose CPR algorithm which has been slightly adapted for the avalanche patient. AvaLife has been designed from scratch, exclusively for this very specific application case. The adapted sequence of actions, inclusion of rescue tactical considerations, advice for cases with multiple burials and multiple patients where there is a shortage of resources, considerations concerning using recovered subjects as a resource in the ongoing rescue, the adapted definition of “injuries incompatible with life”, reasoning behind the importance of rescue breathsas well as the updated BLS iCPR algorithm makes AvaLife the most holistic and most tailor-made tool for avalanche first aid.

In companion rescue and organized rescue with fast intervention times, the decrease in survival chances due to increased burial depth is very important. While the burial depth threshold of ≥150 cm is valid independently of the duration it takes the rescuers to arrive on the scene or the search duration, the difference in the mean number of lives saved is greatest in companion and fast organized rescue. Postponing the excavation of a buried subject at 300 cm burial depth, directly proceeding to the second buried subject where there is a shortage of resources will lead on average to 0.5 additional lives saved compared to excavating the first buried subject independently of burial depth. An advantage of 0.5 lives out of a total of two lives marks a very significant difference in rescue efficiency and the authors hope that this extensive analysis of this long-debated issue proves that the application of a survival chance-optimized burial depth threshold systematically leads to higher overall survival chances and therefore has to be seen as the most and only ethically defendable solution. The likelihood of an air pocket decreases with increasing burial depth, which is key to understanding why mortality increases with increasing burial depth.

Most buried subjects are excavated in less than 60 min and are therefore very high probability normothermic avalanche patients. Where there is a shortage of resources, limit CPR duration to 6 min to provide the “Greatest good for the greatest number” with the limited number of available rescuers.

## Figures and Tables

**Figure 1 ijerph-19-05257-f001:**
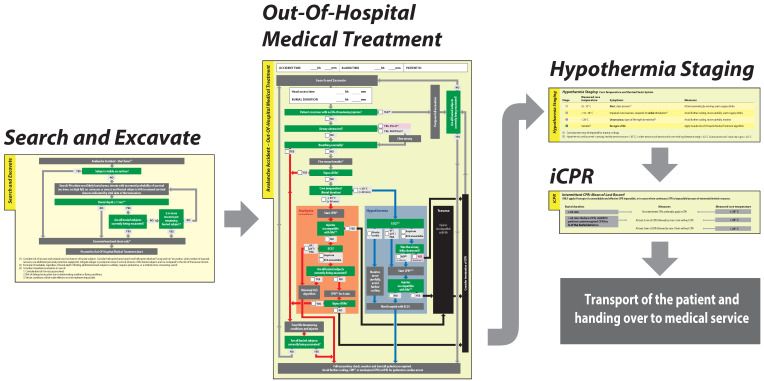
Overview of AvaLife BLS modules. The different AvaLife modules follow step by step the chronology of the rescue and provide critical information for the technical as well as the medical part of avalanche rescue.

**Figure 2 ijerph-19-05257-f002:**
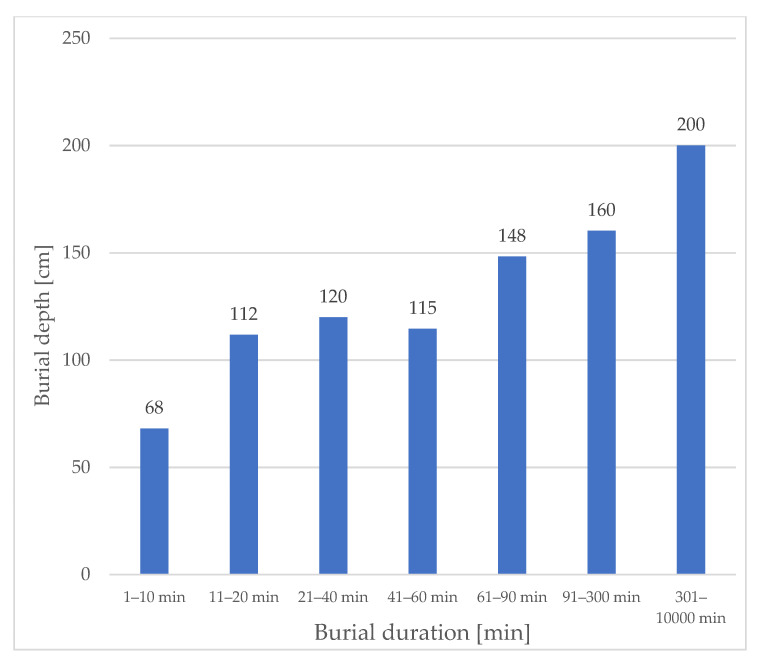
Mean burial depth of excavated subjects for different burial duration ranges. N = 1070.

**Figure 3 ijerph-19-05257-f003:**
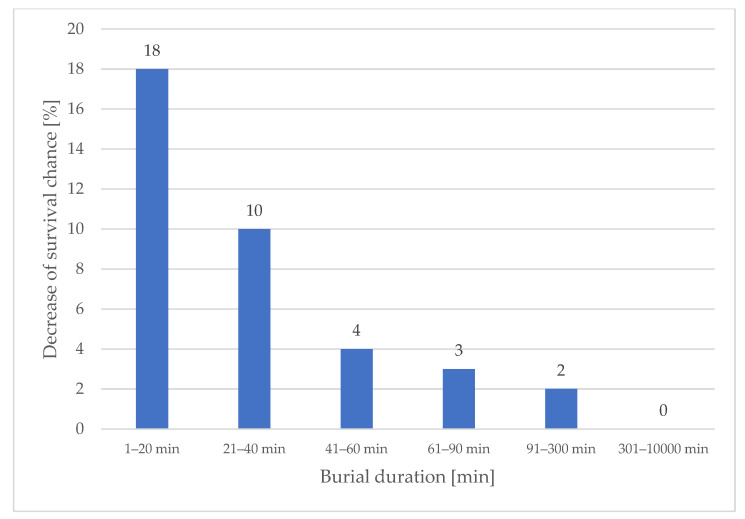
Decrease in survival chances in % per meter of burial depth (N = 1012).

**Figure 4 ijerph-19-05257-f004:**
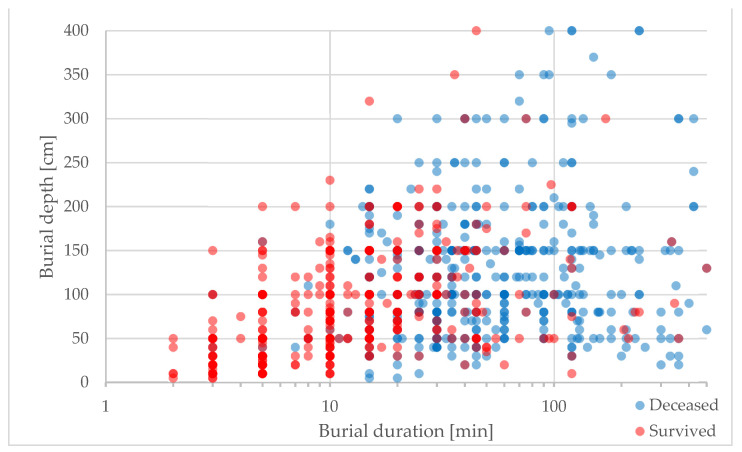
Survival chances based on burial duration and burial depth (N = 1070). The saturation of red for survived and blue for deceased cases increases with the increase in number of cases. For datapoints including survived and deceased cases, the gradually changing shading between red and blue is used to visualize the distribution of survived and deceased cases.

**Figure 5 ijerph-19-05257-f005:**
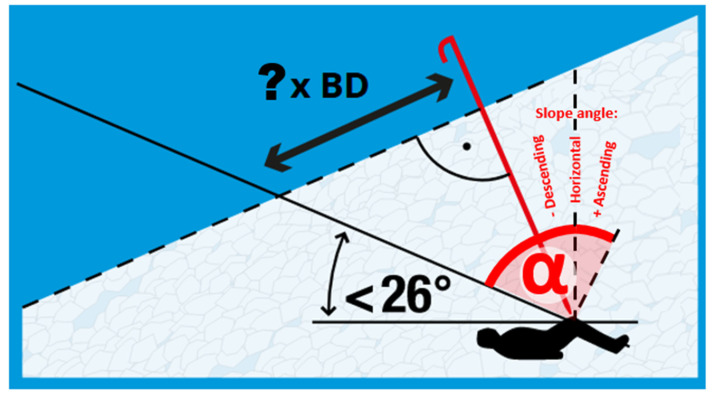
Required length of the snow conveyor belt (tan α) relative to slope inclination. The ramp angle of the snow conveyor belt must remain lower than 26° to prevent the snow to keep falling back towards the probe (empiric field test data [16]) and thus to ensure an efficient excavation effort. Illustration: www.MountainSafety.info as of 31 December 2021, reprinted with permission of www.MountainSafety.info.

**Figure 6 ijerph-19-05257-f006:**
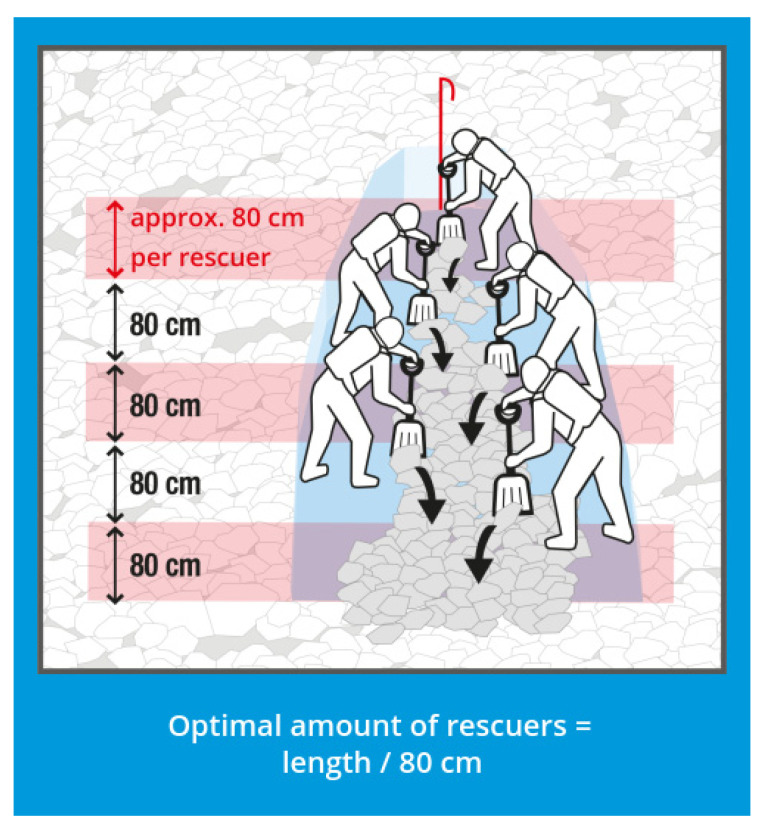
Optimal number of rescuers for efficient excavation applying the snow conveyor belt technique [16]. Illustration: www.MountainSafety.info as of 31 December 2021, reprinted with permission of www.MountainSafety.info.

**Figure 7 ijerph-19-05257-f007:**
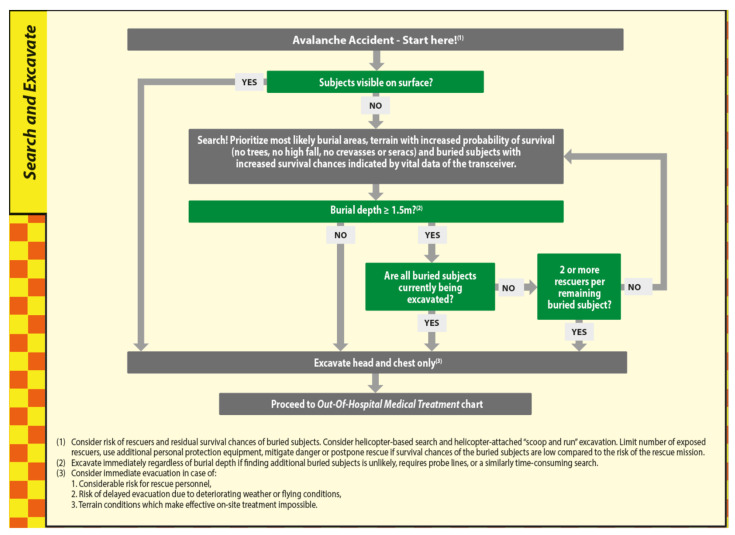
The search and excavate module of AvaLife including the burial depth threshold ≥150 cm and the criterion “2 or more rescuers per remaining buried subject”, which is important to distinguish cases where excavation shall start immediately from cases where the excavation shall be postponed in order to achieve “Greatest good for the greatest number”.

**Figure 8 ijerph-19-05257-f008:**
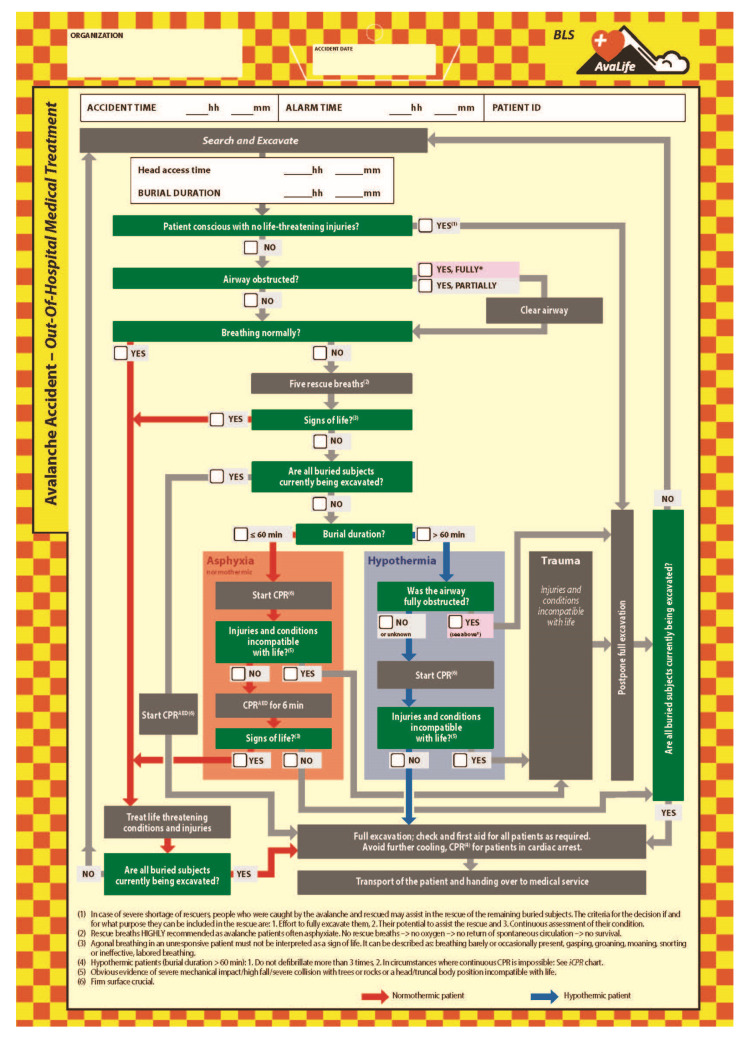
The Out-of-Hospital Medical Treatment module of AvaLife.

**Figure 9 ijerph-19-05257-f009:**
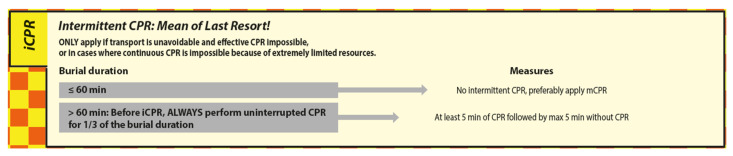
The iCPR module of AvaLife.

**Figure 10 ijerph-19-05257-f010:**
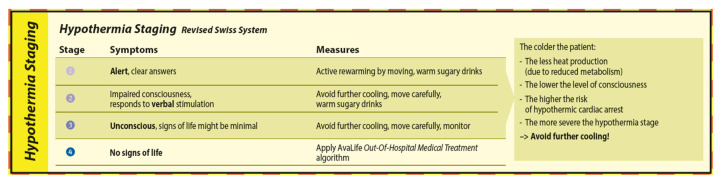
The Hypothermia Staging module of AvaLife.

**Table 1 ijerph-19-05257-t001:** Burial depth threshold and mean number of lives saved.

Search Duration Buried Subject 1 [min]	Medical Treatment Duration Buried Subject 1 [min]	Search Duration Buried Subject 2 [min]	Hard Debris	Soft Debris	
5	10	5	140	140	Survival chance optimized burial depth threshold [cm]
			1.037	1.178	Mean number of lives saved at calculated optimal threshold
			**1.014**	**1.159**	**Mean number of lives saved at 150 cm threshold**
			0.53	0.67	Mean number of lives saved if first buried subject gets excavated in 300 cm burial depth
			0.48	0.49	Additional lives saved if burial depth triage criterion applied instead of excavating first buried subject at 300 cm
10	10	5	150	150	Survival chance optimized burial depth threshold [cm]
			0.904	1.001	Mean number of lives saved at calculated optimal threshold
			**0.904**	**1.001**	**Mean number of lives saved at 150 cm threshold**
			0.40	0.70	Mean number of lives saved if first buried subject gets excavated in 300 cm burial depth
			0.50	0.30	Additional lives saved if burial depth triage criterion applied instead of excavating first buried subject at 300 cm
25	10	5	110	160	Survival chance optimized burial depth threshold [cm]
			0.590	0.623	Mean number of lives saved at calculated optimal threshold
			**0.588**	**0.623**	**Mean number of lives saved at 150 cm threshold**
			0.44	0.35	Mean number of lives saved if first buried subject gets excavated in 300 cm burial depth
			0.15	0.27	Additional lives saved if burial depth triage criterion applied instead of excavating first buried subject at 300 cm
40	10	10	160	180	Survival chance optimized burial depth threshold [cm]
			0.475	0.492	Mean number of lives saved at calculated optimal threshold
			**0.473**	**0.491**	**Mean number of lives saved at 150 cm threshold**
			0.41	0.41	Mean number of lives saved if first buried subject gets excavated in 300 cm burial depth
			0.06	0.08	Additional lives saved if burial depth triage criterion applied instead of excavating first buried subject at 300 cm

## Data Availability

Data used in this paper is either proprietary and/or protected by the privacy protection act or other, similar rules and regulations and therefore not publicly accessible.

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
