# Peer review of "AvaLife—A New Multi-Disciplinary Approach Supported by Accident and Field Test Data to Optimize Survival Chances in Rescue and First Aid of Avalanche Patients"

_ijerph, 2022, doi:10.3390/ijerph19095257_

Round 1
Reviewer 1 Report
The paper presents a new method to save the avalanche patients. The method is useful and well supported by numerical evidence. The simulation needs some improvement by using some more values as far as Table 1 is concerned. It would be suitable if some statistical model is fitted to see the effect of burial duration and burial time on the survival of the victim.
Author Response
Dear Reviewer,
Thank you for your much appreciated review effort and your valuable comments!
We would like to answer the specific points you have raised as follows:
1:
The simulation needs some improvement by using some more values as far as Table 1 is concerned.
Due to page size and font size limitations, we were only able to include some of the parameters in the overview table 1. However, it is important to understand that the simulation in fact includes more input values such as:
- probability of survival from the Haegeli et al. survival curve
- randomly picked burial depth based on Swiss avalanche accident data for buried subject 1 and 2.
- the required patient treatment time
The additional variables are mentioned in the „methods“ section of the paper.
2:
It would be suitable if some statistical model is fitted to see the effect of burial duration and burial time on the survival of the victim.
Concerning the analysis of the burial depth threshold in case of shortage of resources, we have concluded that an approach based on a statistical model, or a linear or non-linear equation, is unlikely to be successful as the complexity of the interaction between the many variables seems simply to high for any satisfactory fitting. This conclusion is in fact the reason why we consider a Monte Carlo simulation as the only feasible approach to the problem.
However, burial duration (as well as burial depth) is an input variable of the Monte Carlo simulation and therefore their effect is fully considered in the outcome of the simulation. Looking at the impact of burial duration and burial depth to survival of a single buried subject, please consult figure 3 as well as the Haegeli et al. survival curve.
Best regards,
AvaLife Authors and Co-Authors
Reviewer 2 Report
Referee report on “AvaLife – A New Multi-Disciplinary Approach Supported by 2 Accident and Field Test Data to Optimize Survival Chances in 3 Rescue and First Aid of Avalanche Patients” submitted to IJERPH
The work performs an analysis of the AvaLife algorithm, combining sample information and Monte Carlo simulation studies, and based on these results suggests some optimal strategies for rescue procedures to Optimize Survival Chances in 3 Rescue and First Aid of Avalanche Patients.
The purpose of the work is quite clear, the authors demonstrate a mastery of the problem at hand, and the work is well motivated and of great practical importance. So I see a lot of merit in the work submitted.
However, I believe that the work needs to improve the method description part. There is no clear description of the Monte Carlo algorithm used, nor how the parameters of the simulation procedures are estimated/imputed. It is unclear whether it is based on sample means or whether duration curves are estimated using some statistical model to estimate duration models. So my recommendation is to introduce a section describing the simulation and parameter estimation algorithms used in the analysis.
Also at times it is difficult to understand the source for some statements, if they are based on simulation results, if they are results of the sample used, etc. For example in Lines 127-132 'None of the existing survival curves for fully buried subjects differentiated be- 127 tween different burial depths; therefore, the probability of survival for any given burial 128 duration always includes buried subjects at different burial depths…”
Is it a result of the analysis? Is it a result of other studies? Is it a result of the simulations?
So even if the work is well founded, I would like to see a section carefully describing the analysis methodology used, as well as the parameters estimation/calibration procedures used in the Monte Carlo study, and also a better separation of what are the results of the study or analysis assumptions. It is a very interesting and useful work, but the methodological part needs more details and a better motivation of the results based on the results of the analysis methodology. So my recommendation is for a revision of the work, based on these two main suggestions.
Author Response
Dear Reviewer,
Thank you for your much appreciated review effort and your valuable comments!
We would like to answer the specific points you have raised as follows:
1:
However, I believe that the work needs to improve the method description part. There is no clear description of the Monte Carlo algorithm used, nor how the parameters of the simulation procedures are estimated/imputed. It is unclear whether it is based on sample means or whether duration curves are estimated using some statistical model to estimate duration models. So my recommendation is to introduce a section describing the simulation and parameter estimation algorithms used in the analysis.
Based on random sampling within the Swiss accident database:
- burial depth of buried subject 1 and 2
Based on static parameters:
- Search time 1 and search time 2 (see additional reasoning behind these values in the extended methods description)
- medical treatment time
- excavation performance for hard and soft debris
Based on Haegeli et al. survival curve:
- Probability of survival at burial duration typical burial depth based on figure 2
Based on figure 3:
- Burial depth adapted survival chance
The “methods” chapter specifically makes reference to the input parameters as listed above.
2:
Also at times it is difficult to understand the source for some statements, if they are based on simulation results, if they are results of the sample used, etc. For example in Lines 127-132
Used source data now explicitly mentioned.
3:
'None of the existing survival curves for fully buried subjects differentiated be- 127 tween different burial depths; therefore, the probability of survival for any given burial 128 duration always includes buried subjects at different burial depths…”
Is it a result of the analysis? Is it a result of other studies? Is it a result of the simulations?
This is simply stating the fact that the Haegeli et al. survival curve only shows survival/non-survival in function of burial duration, independently of the burial depth of the buried subjects.
4:
So even if the work is well founded, I would like to see a section carefully describing the analysis methodology used, as well as the parameters estimation/calibration procedures used in the Monte Carlo study, and also a better separation of what are the results of the study or analysis assumptions
See additional clarifications added to the paper and above.
There is otherwise no “calibration” procedure possible, needed or done for the MC simulation or the input variables, which in fact is a major advantage of the MC approach, it simply takes to raw facts and allows to test the impact of specific procedures (such as the impact of different burial depth thresholds to survival chances).
Best regards,
AvaLife Authors and Co-Authors
Reviewer 3 Report
First of all the lecture on the paper was interesting. However, I would like to make some comments and remarks:
1) introduction, the sentence "Snow sports in the backcountry have seen a steep increase in popularity. In North America and Europe, almost 200 backcountry users die in avalanches every year." I think a citation (reference) could be useful.
2) Can anyone (any country, company) apply the proposed approach, or is a patent/permit, etc. is needed?
3) Fig. 1 (Overview of AvaLife BLS modules) is hard to read. In my opinion, a short description could be beneficial.
4) Did the authors try to obtain avalanche data from other countries?
5) As a statistician, I would like to see the estimated mean intervals for the presented data.
6) As I'm not a rescue expert I would like to know how simulated time frames reflect reality.
7) Can the same idea/method be used in the case of earthquakes, mudflows that can burry people too?
8) In the case of the results of the analysis I think standard deviations should be added so it would be easier to interpret the results.
Author Response
Dear Reviewer,
Thank you for your much appreciated review effort and your valuable comments!
We would like to answer the specific points you have raised as follows:
1:
introduction, the sentence "Snow sports in the backcountry have seen a steep increase in popularity. In North America and Europe, almost 200 backcountry users die in avalanches every year." I think a citation (reference) could be useful.
Numbers removed since data is not systematically collected anymore in the recent past and absolute numbers are irrelevant.
2:
Can anyone (any country, company) apply the proposed approach, or is a patent/permit, etc. is needed?
Available for free use by everyone in 20+ languages, see www.MountainSafety.info
3:
Fig. 1 (Overview of AvaLife BLS modules) is hard to read. In my opinion, a short description could be beneficial.
All excerpts of the algorithm now provided as eps data to the editor for professional final layout work.
4:
Did the authors try to obtain avalanche data from other countries?
No, because FR, AT and CH have the best possible ratio between amount and quality of available data.
5:
As I'm not a rescue expert I would like to know how simulated time frames reflect reality.
See newly inserted text.
6:
Can the same idea/method be used in the case of earthquakes, mudflows that can burry people too?
For earthquakes yes, very much indeed! See collaborative approach outlined in:
Genswein, M.; Thorvaldsdóttir, S.; Zweifel, B. In Remote Reverse Triage in Avalanche Rescue. , Proceedings of the International Snow Science Workshop, Whistler, Canada, September 21-27 2008, 2008; Whistler, Canada, 2008.
Best regards,
AvaLife Authors and Co-Authors
Round 2
Reviewer 2 Report
Overall I am satisfied with the changes to the article. I believe it follows the presentation standards in this literature, and on the whole I see it as having merit for publication in its present form.